# Academic research values: Conceptualization and initial steps of scale development

**Andrea Kis**, **Elena M. Tur, Krist Vaesen, Wybo Houkes, Daniël Lakens**

Department of Industrial Engineering & Innovation Sciences, Eindhoven University of Technology, Eindhoven, Netherlands

* a.kis@tue.nl

## Abstract

We draw on value theory in social psychology to conceptualize the range of motives that can influence researchers' attitudes, decisions, and actions. To conceptualize academic research values, we integrate theoretical insights from the literature on personal, work, and scientific work values, as well as the outcome of interviews and a survey among 255 participants about values relating to academic research. Finally, we propose a total of 246 academic research value items spread over 11 dimensions and 34 sub-themes. We relate our conceptualization and proposed items to existing work and provide recommendations for future scale development. Gaining a better understanding of researchers' different values can improve careers in science, attract a more diverse range of people to enter science, and elucidate some of the mechanisms that lead to both exemplary and questionable scientific practices.

## 1. Introduction

Growing interest in empirically studying the scientific process (i.e., research on research, or meta-science) has increased the focus on the psychological constructs underlying researchers' attitudes, decisions, and behaviors. Studies on a range of constructs such as personality traits [1], attitudes [2], career incentives [3], values [4], and motivations [5] have examined why researchers do what they do. Our conceptual paper contributes to this growing body of research by integrating and extending existing scientific work values.

Understanding values specific to scientific work helps to make scientific careers more attractive for a more diverse group of scholars. Extrapolating a long line of research studies documenting the influence of values on behavior [6,7] can provide insights into why some researchers conduct exemplary work, yet others engage in questionable scientific practices. An improved conceptualization and measure of values specific to the academic context could enhance our empirical understanding of the role that values play in research practices, as a way of assessing the outcomes of responsible conduct of research courses, and as a tool for exploring the personal differences between researchers of various nationalities, disciplines, and at various stages in their careers [4].

To predict research-related behaviors, we aimed to develop a scale to measure researchers' psychological values. While values are broadly used constructs across various disciplines, our approach focuses specifically on psychological values. The current study is rooted in research

**Data availability statement:** All files are available in a public OSF repository: Kis, A.

(Creator) (24 Apr 2023). What do researchers value? OSF. DOI 10.17605/OSF.IO/ESJC2

**Funding:** DL was funded by Vidi Grant 452-17-013 from the Dutch Research Council (NWO). The funders had no role in study design, data collection and analysis, decision to publish, or preparation of the manuscript.

**Competing interests:** The authors have declared that no competing interests exist.

that defines values as underlying psychological criteria guiding behaviors and preferences. This focus is due to our goal of describing individuals' values and connected behaviors, necessitating a construct capable of achieving this. Psychological values are widely acknowledged for their ability to provide a descriptive, rather than normative, approach to understanding people's motivations and goals. Consequently, our approach excludes broader definitions of values, such as core universal moral values [8], scientific virtues [9], and scientific values [10].

In the remainder of this paper, we present these scale development steps. First, to ground our work, we discuss the personal, work, and scientific work value literature. Next, we present our methods, starting with the concept development steps that serve as precursors of developing a conceptually sound, comprehensive description of psychological values relevant to researchers in a scientific context. Then, we highlight the item generation process through which we delimited and finalized our definitions and through which we created an initial set of scientific work values that can form the basis for future measure development work (see the process summarized in Fig 1).

## 2. Literature review

### 2.1. What are values?

Values are considered central to human behavior [11,12]. As such, they serve as an essential, unifying construct to many fields in the humanities, social sciences, and especially psychology [6]. Two lines of research are especially relevant for studying scientific work values: personal values and work values. While personal values can guide researchers in everyday situations, researchers operate in an environment characterized by sets of rules, norms, and reward structures specific to academia; their work values are therefore equally important for conceptualizing scientific values. However, whereas work values recognized in the literature such as ambition or having challenging work are easily applicable to research, other work values need to be adapted to the scientific context. Academia has a specific culture and reward structure, and we should expect a different set of relevant work values compared to non-academic jobs.

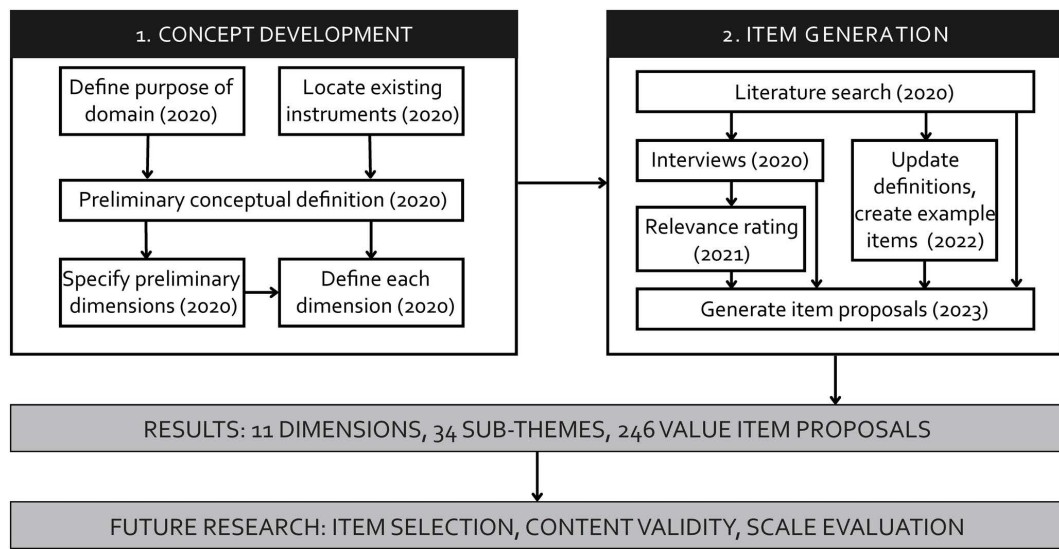

**Fig 1. Overview of the scale development process, results, and future research (in brackets: the year of carrying out the activity).**

*Personal values* are desirable goals that vary in importance, transcend specific situations, and serve as guiding principles in the life of a person or group [13,14]. They are the key determinants of a wide range of beliefs and behaviors [7]. The most prominent and well-validated personal value theory by S. H. Schwartz [13] maps two dimensions of unique underlying values (Fig 2).

The first dimension contrasts "openness to change" and "conservation" values, thus capturing a conflict between valuing independence, thought, and readiness to change, versus an appreciation of order, self-restriction, and resistance to change. The second dimension ranges from "self-enhancement" to "self-transcendence" and captures self-interest and relative success concerns versus the interests and welfare of others. Self-enhancement values are positively associated with unethical behaviors and competition, while self-transcendence values facilitate cooperation and prosocial behavior [7]. Values can be further subdivided into ten motivational types labeled stimulation, self-direction, universalism, benevolence, tradition, conformity, security, power, achievement, and hedonism. Studies show that these categories maintain their relevance across cultures and provide a comprehensive, universal measure of human values [6,15]. A study by Knafo and Sagiv [16] comparing researchers' personal values to those in other professions concluded that investigative occupations (including science) emphasize self-direction values and to a lesser extent benevolence and universalism, but attribute low importance to hedonism, tradition, and security values.

A second line of relevant research concerns *work values.* They comprise a specific, distinguishable set of values relating to a person's working life, that can predict or are linked to a wide range of work-related attitudes, behaviors, and outcomes, such as work satisfaction, career choice, work-related decisions, career adaptability, work engagement, motivation, basic work-related psychological needs, work-related learning, and life satisfaction (as summarized by Busque-Carrier et al [17]).

The work values literature notably lacks an individual, unifying, widely accepted, and comprehensive definition and delimitation of work values. The fragmented nature of the work values literature [18], as well as the conceptual confusion [19] can hinder the progress of work value research [17,20,21]. Recent efforts to better understand work values have yielded construct validity studies based on widely used measures of work values [22], connected the parallel paths of work and personal values by customizing the motivational types underlying

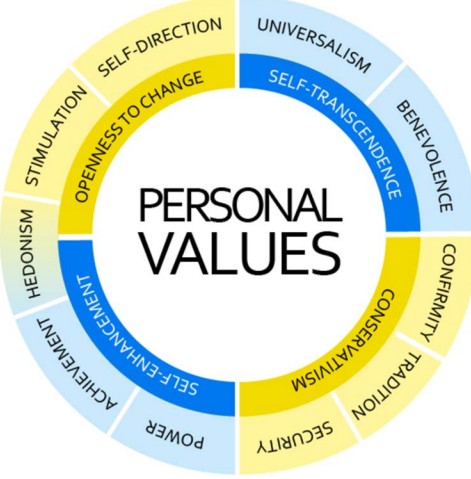

**Fig 2. Structure of value relationships.** Redrawn based on S.H. Schwartz [13].

personal values (as posited by S. H. Schwartz [23]) to the work context [24], and developed a valid work values scale based on a comprehensive literature review [17]. These lines of thought cluster work values into six [22], eleven [24], or fifteen [17] components, as depicted in Fig 3.

## 2.2. Scientific work values

In the past twenty years, few empirically driven psychological approaches have identified values relating to scientific work. Prior studies aimed to compare various professions' personal and work value profiles [16], utilized value measures to study sub-populations of researchers such as describing economists' personal values [25], or examined gender differences in the impact of work values on research careers [26]. We know of two studies that pursue a psychological approach to scientific values. One is by Demirutku and Güngör [27], who add a "scientific values" label to the circular structure of S. H. Schwartz's personal value theory [13]. However, their approach adds value items drawn from sociological and educational sources not in line with the psychological conceptualization of values (e.g., rationality or objectivity) as a set of personal value items without further integration.

The other study presents a measure of work values specific to scientific research. The Values in Scientific Work scale, proposed by English et al [4], is the first such instrument based on a psychological conceptualization of work values and the research integrity literature. This scale comprises eight subscales: autonomy, research ethics, social impact, income, collaboration, innovation and growth, preserving relationships, and job security. Despite being a much-needed step towards conceptualizing values in scientific work, the Values in Scientific Work

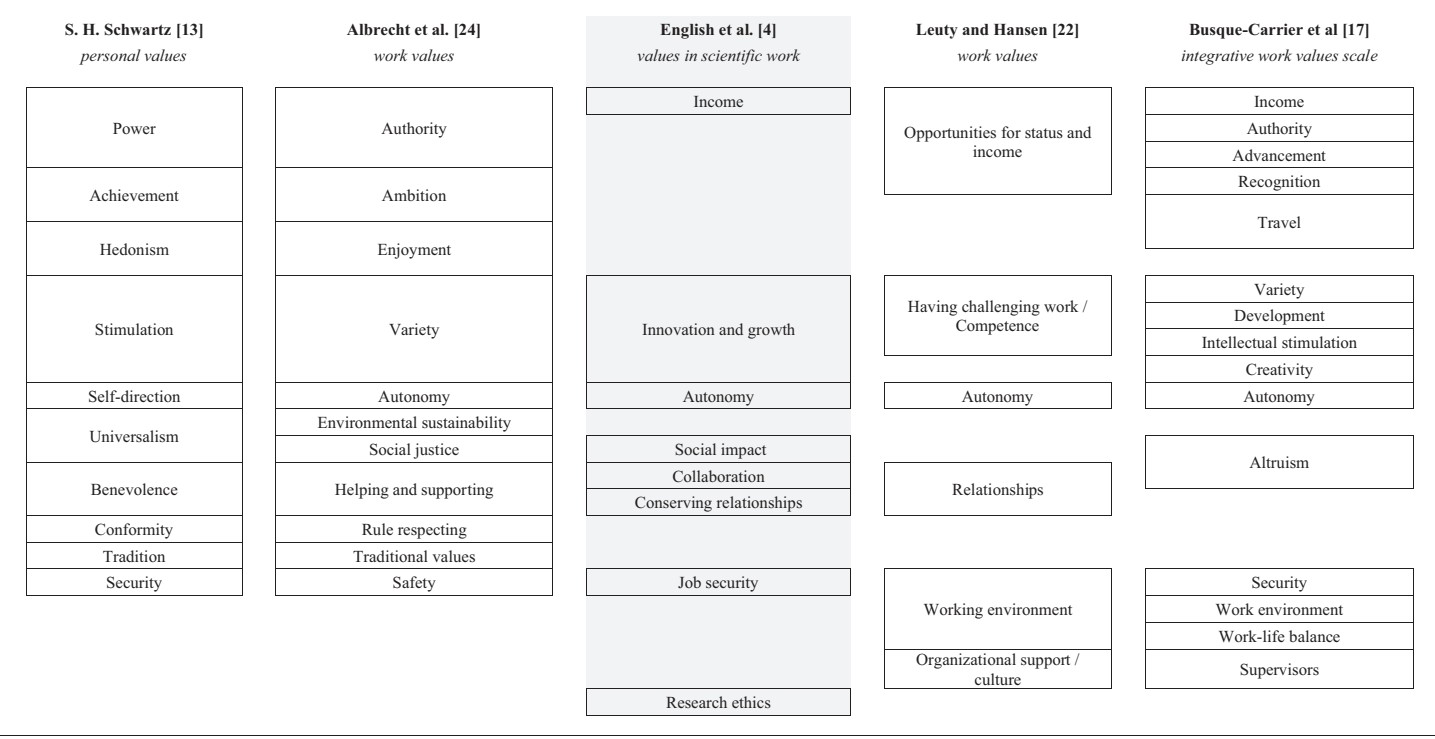

**Fig 3. Comparison of main value categories based on value domain descriptions by S. H.** Schwartz [13], *Albrecht et al* [24], *English et al* [4], *Leuty and Hansen* [22], *and Busque-Carrier et al* [17].

scale does not incorporate several of the value dimensions commonly found in the literature (Fig 3). Since our approach seems to mainly align with how the Values in Scientific Work scale is constructed, it is worthwhile to specify these gaps.

First, regarding personal values, the Values in Scientific Work scale does not include items relating to achievement, hedonism, conformity, and tradition. Having a social impact as reflected in items such as "It is important to her/him that research helps to solve real world problems" might be a relevant value to researchers and does capture some of the universalism construct posited by S. H. Schwartz [13]. However, building on the conceptualization of personal values embedded in a work setting by Albrecht et al [24], who conceptualize universalism as including values emphasizing the importance of environmental sustainability and social justice, universalism in the context of scientific work values could be conceptualized as a broader construct that reflects a need for understanding, appreciation, tolerance, and protection for the welfare of all people and of nature.

English et al's [4] important work could be naturally extended with a wider set of power values in line with S. H. Schwartz [23], that reflect the need for social status and prestige, control or dominance over people and resources, and include values such as wealth and authority. With items such as "Earning a good salary is important to her/him," English et al's [4] power-related values successfully incorporated the wealth or income aspect from this definition. Other natural extensions would be elements of power values potentially relevant to studying researchers' behaviors and decisions. For example, status is an often cited motivation for researchers [e.g., 5,28,29]. Excluding such features might reduce the instrument's measuring potential.

Second, compared to the core work values that Leuty and Hansen [22] outline, the Values in Scientific Work scale does not include items reflecting the importance of working environments: the physical conditions, quality of management or supervision, job security, supportive co-workers, work-life balance, and organizational support (i.e., not being pressed to engage in work that can be seen as immoral), management support, proper training, fair organizational policies, and clear procedures. Opportunities for status and income are only partly covered by items related to income. The Integrative Work Values Scale recently developed by Busque-Carrier et al [17] yields comparable results; it contains value categories pertaining to authority, recognition, travel, work environments, work-life balance, and supervisors, unlike the Values in Scientific Work scale.

Even though features such as income are aligned with the work values posited by Leuty and Hansen [22] and Busque-Carrier et al [17], the wording in the Values in Scientific Work scale may be a limited interpretation of what wealth and resources mean within the (scientific) work environment. In S. H. Schwartz's [13] original construct, valuing wealth indicated the need for control over resources and for prestige, a notion mirrored in two of Albrecht et al's [24] work value items: "To have authority over limited resources," and "To determine how money is spent." While items in the Values in Scientific Work scale such as earning a high salary could be relevant to researchers, having control over research resources might be more important. For example, in a study on researchers' motivations, Lam [5] found that while the importance of increasing funding and research resources scored top of the motivational hierarchy, increasing personal income was at the bottom.

In conclusion, we regard the Values in Scientific Work scale as a key step towards an overview of important work values in science. However, in our view, the metascience and psychology of science fields need a more comprehensive approach that systematically extends and integrates the diverse insights in the literature, as shown in Fig 3. This extension is important to achieve a complete set of values for studying researchers' actions, attitudes, and beliefs.

## 3. Methods

### 3.1. Measure development: Process overview

Compiling a scale is a complex, iterative process, often involving a wide variety of qualitative and quantitative methods. Boateng et al [30] distinguish three phases: item development, scale development, and scale evaluation. These steps are listed and explained in Table 1 (columns 1 and 2). Scale construction guidelines tend to emphasize the importance of the initial steps to generate items, yet practical examples and advice on these steps are often lacking [35]. To document transparently and solidify our understanding of the construct and its domains, and following the example of authors who rigorously discuss these initial steps [35–39], we focus exclusively on the first of the three phases: item development.

### 3.2. Domain identification and item development.

We first identified and defined the domain of interest based on literature reviews and discussions. Having reviewed the values literature to find the most widely utilized conceptualizations and instruments, we arrived at a preliminary conceptual definition and specification of the scientific work value dimensions. Then, during the generation phase, we prepared an initial set of items to assess later for content validity by integrating insights from a mix of deductive and inductive methods — researchers' values obtained from a literature review, interviews, and a survey. Before continuing, we updated our preliminary definitions based on the conclusions from these methods. We then generated the set of item proposals that will be presented as our main result. This item generation process and an overview of our methods and results are detailed in Table 1 and the following section.

## 4. Results

### 4.1. Concept development

This study aimed to identify scientific values that can serve as determinants of researchers' scientific actions. Our initial conceptualization of relevant instruments and provisional definition of the concept formed the basis for an iterative process of concept development [40], initially mainly influenced by S. H. Schwartz's [13] original value framework. At this stage, we did not rule out that work values or scientific work values would be a more suitable basis, but decided to progress from generic to specific values, following recent integrative approaches that grounded their work in Schwartz's personal value theory.

Accordingly, we defined scientific work values as "desirable goals or motivators within the scientific context that transcend specific situations, vary in importance, and serve as guiding principles for a researcher or group of researchers' science-related decisions." With our focus on scientific research in an academic setting and related generic tasks and goals, we purposely excluded values concerning activities usually linked to academic positions, such as teaching and administration, to limit our study's scope. Definitions of dimensions were modified to the scientific work context based on the original definitions (Table 2).

### 4.2. Item generation

To identify appropriate questions for each domain, we combined the deductive methods based on the literature review with existing measures and inductive methods informed by the responses in personal interviews and an online survey.

**4.2.1. Literature search.** S. H. Schwartz's theory is the predominant and empirically best-validated assessment of personal values in surveys [41]. In line with the conceptual definition and underlying theoretical assumptions, we based our initial set of items on S. H. Schwartz's [13] Value Survey (SVS, 57 items).

**Table 1. Step-by-step overview of the process to develop the Academic Research Values (ARV) scale.**

| Steps | Aim | Methods | | Results |
|---|---|---|---|---|
| **1. Concept development: Specify the boundaries of values in science and facilitate item generation** | | | | |
| **1.1 Define purpose of domain** | Identify the domain and purpose of the construct to be measured | Literature search | | Domain of interest: Scientific values. Purpose: Find scientific values that can serve as determinants to scientific actions of researchers using S. H. Schwartz's personal values as a basis. |
| **1.2 Locate existing instruments** | Identify measuring instruments for studying scientific values, with a specific focus on Schwartz's personal values | | | A list of instruments measuring personal values. We excluded other sources, such as moral values and scientific virtues, but included studies utilizing value measures potentially relevant for researchers (see list at 2.1). At this point, we tested whether personal values are the best basis in terms of constructs and phrasing (i.e., not too generic, or abstract for our purposes), see outcomes at 2.2. |
| **1.3 Preliminary conceptual definition** | Provide a preliminary conceptual definition of scientific values | Team discussion | | "Scientific values are desirable goals or motivators within the scientific context that transcend specific situations, vary in importance, and serve as guiding principles for researchers' science-related decisions. Scientific values relate to research activities, generic tasks and goals, but exclude specific values of activities connected with academic positions, such as teaching and administration." |
| **1.4 Specify preliminary domain dimensions** | Specify the dimensions in scientific values | Literature search | Team discussion | Preliminary dimensions match dimensions specified by S. H. Schwartz [13]: Self-direction, Stimulation, Hedonism, Achievement, Power, Security, Conformity, Tradition, Benevolence, Universalism (Table 2). We assumed that additional dimension(s) specific to the scientific context might be present, but before including new dimensions or items we wanted to check if personal values can serve as a solid basis for scientific values (as described in our inductive methods at 2.2). |
| **1.5 Define each dimension** | To provide a preliminary definition and delimitation of each dimension of scientific values | Team discussion | | Preliminary dimensions coincide with dimensions specified by S. H. Schwartz [13]. Definitions were modified to the scientific context (see Table 2). |
| **2. Item generation: To identify appropriate questions that fit the identified domain** | | | | |
| **2.1 Deductive methods** | To collect data from the literature | Literature search for possibly relevant value items in validated value instruments | | **Schwartz's personal values**: S. H. Schwartz's Value Survey (SVS) - 57 items [31] <br> **Work values**: Values at Work (VaW) - 52 items [24], Minnesota Importance Questionnaire (MIQ) - 20 items [32], Super's Work Values Inventory - Revised (SWVI-R) - 12 items [33], Manhardt's Work Values Inventory (MWVI) - 25 items [34] <br> **Scientific values**: Values in Scientific Work (VSW) - 35 items [4] <br> (Value items and dimensions from these scales are listed in S1 Appendix and S2 Appendix 2) |
| **2.2 Inductive methods** | To collect data from samples within the target group | Interviews N = 6 | Relevance rating N = 255 | We used the **Schwartz Value Survey 57 items** as an initial basis for the interviews and a modified version (**60 items**) for the relevance rating survey. Based on our outcomes, short formats of personal values seem less relevant to the scientific context. Accordingly, we revisited the work values literature and modified our approach. (S3 Appendix) |

*(Continued)*

**Table 1.** (Continued)

| Steps | Aim | Methods | Results |
|---|---|---|---|
| **2.3 Update definitions, generating the item proposals** | To update the definitions and delimitations of the construct, dimensions, and items. Reevaluating possibly relevant concepts. | Team discussion | Summarizing all sources reviewed so far, we redefined our construct as: "**Academic research values** are principles which serve as a basis of evaluating outcomes of scientific work-related actions, guide the selection of scientific work goals, and represent the relative importance assigned to various academic job aspects related to research activities. They serve as guiding principles in the decisions of a researcher or groups of researchers in the academic work setting and are less broad than personal values, but still represent motivational goals that transcend specific work situations. Similar to personal values, academic research values are desirable in the sense that they represent important and worthy causes to researchers."<br>We made decisions about item characteristics (i.e., wording, form, and response types), generated an initial set of items (11 dimensions, 34 sub-themes, 246 items), and compared our items to the Integrative Work Values Scale as a control. (S3 Appendix - item development, S4 Appendix - dimension development, S5 Appendix - item proposals, S6 Appendix - comparison) |

**3. Item selection and content validity: To assess if the generated items adequately measure the domain of interest—*Future research***

A literature search conducted from January to May 2020 identified instruments that measure relevant values for scientific work. We sought reviews of validated measures of work values using these search terms: work value combined with scale, measur*, instrument, inventory, questionnaire, model, validity or validat* in the Web of Science, APA PsycArticles, APA PsycInfo, ERIC, and SCOPUS databases. Given the scope and aim of the review, no restrictions were placed on the publication period. The search yielded a substantial number of articles (3027 results in total, 1941 results after excluding duplicates), which were then screened for relevance based on titles and abstracts. Full-text articles were reviewed to determine their alignment with the research objectives. We only included studies conceptualizing work values as psychological constructs and excluded constructs not in line with our preliminary value definition, for example, social work values, occupational work values, and work ethic values (i.e., Confucian, Islamic, protestant). Because synthesizing the large number of different conceptualizations of work values was beyond the scope of our research, we further limited our search to sources that provide summaries or comparisons of existing and widely used scales.

We reviewed four such sources, described below. Due to the limitations in three studies, we only included Leuty and Hansen's [22] examination of overarching work values in three instruments. These authors found six common work value factors through exploratory factor analysis: working environment, competence/having challenging work, opportunities for status and income, autonomy, organizational support/culture, and relationships. Earlier studies [42,43] compared several measures to demonstrate that some values were systematically captured across many instruments, but relied on anecdotal information rather than empirical validation [22]. Macnab and Fitzsimmons [44] addressed this limitation, however their findings are 35 years old and might need to be replicated, especially in light of critique about their usefulness and generalizability. As Leuty and Hansen [22] point out, some of the scales Macnab and Fitzsimmons used have become less relevant over time, their sampling was non-representative in their study, and further content validation is lacking.

During our search for work value instruments, we found that Schwartz's value instruments were cited as either work value measures [e.g., 38] or as basis for other developed work value measures [e.g., 24,45–47]. We included the 11-factor work value model [24 and see Fig 3] that addressed several limitations in earlier personal value-based research.

We also included the only validated measure of scientific work values known to us, centered primarily on work values [4] to supplement instruments based on personal values.

**4.2.2. Interviews.** To understand how researchers interpret personal value items in science, we set up two consecutive studies: a qualitative, interview-based investigation to establish researchers' understanding of value items; and a quantitative survey to determine what relevance researchers assign to different versions of value items.

We conducted semi-structured interviews through 16-3-2020 to 30-4-2020 to study researchers' understanding of generic value items. Our primary aims were to see if the wording, clarity, and form of items relating to personal values are adequate; to check whether some value items might be too ambiguous or irrelevant for our purposes; and to identify any values that might be relevant to researchers but are not represented in the Schwartz Value Survey set of personal value items. The project was registered in the study proposal and ethical form approved by the Eindhoven University of Technology (TU/e) Ethical Review Board (ID: 1074, see project's OSF repository).

We recruited six participants. The sample size estimation was based on feasibility considerations and methodological recommendations on saturation in qualitative studies [48,49]. As our initial research questions were generic at this stage and we intended to follow up any results with other research steps, we determined the sample size based on the lowest saturation point estimate of six participants presented in several qualitative methodology literature

**Table 2. Definitions of preliminary dimensions based on Schwartz's motivational types [13].**

| Dimensions | Original definitions [13] | Reframed definitions |
|---|---|---|
| Self-direction | Independent thought and action: choosing, creating, and exploring (freedom, creativity, independent, choosing my own goals, curiosity) | Freedom of thought and action: determination of research tasks, creating, and exploring own research topics |
| Stimulation | Excitement, novelty, and challenge in life (exciting life, varied life, daring) | Being drawn to excitement, variety, novelty, and challenge in research |
| Hedonism | Pleasure and sensuous gratification for oneself (pleasure, enjoying life, self-indulgent) | Seeking to take pleasure and gratification within the realm of research |
| Achievement | Personal success through demonstrating competence according to social standards (ambitious, capable, influential, successful) | Scientific success through demonstrating competence according to academic standards, feelings of achievement and being a competent researcher |
| Power | Social status and prestige, control or dominance over people and resources (social power, wealth, authority) | Scientific status and prestige, control or dominance over other researchers and research resources |
| Security | Safety, harmony and stability of society, relationships, and self (social order, national security, family security, reciprocation of favors, clean) | Safety within the research environment |
| Conformity | The restraint of actions, inclinations, and impulses that are likely to upset or harm others and violate social expectations or norms (politeness, self-discipline, respect for elders, obedient) | Conformity to scientific norms, restraint of actions that might upset or harm others, abiding by social norms within the research environment |
| Tradition | Respect, commitment and acceptance of the customs and ideas that traditional culture or religion provides (respect for tradition, modest, humble, accepting my portion in life, devout) | Modesty about achievements and role in forming science, respect, and acceptance of scientific traditions |
| Benevolence | Preservation and enhancement of the welfare of people with whom one is in frequent personal contact (loyal, responsible, honest, helpful, forgiving) | Preservation and enhancement of the welfare of colleagues |
| Universalism | Understanding, appreciation, tolerance, and protection for the welfare of all people and of nature (equality, unity with nature, wisdom, world of peace, world of beauty, social justice, broad-minded, protecting the environment) | Understanding, tolerance, and appreciation of socially relevant issues, sense of need to contribute to sustainability and social research |

sources [50]. We used a convenience sample consisting of six PhD candidates (university employees in the Netherlands, four internationals and two Dutch nationals, four women and two men, five in their first PhD year and one in their final year).

The interviewees provided written informed consent and then were asked about their impressions of the 57 values listed in the Schwartz Value Survey (S1 Appendix). Some deemed values like cleanliness difficult to understand in the context of researchers' motivations. Values such as reciprocation of favors, respect for tradition, mature love, detachment, and unity with nature, were deemed completely irrelevant. The interviewees reflected on how most values were easy to understand, however, many ambiguities in interpretation surfaced. For example, even though the interviewees felt wealth was relevant, several commented on how spiritual matters are more important to researchers and why wealth is irrelevant or at least a bizarre motivation to do science. Some reflected on a different aspect of wealth: financial stability. We also found ambiguities in interpretations of other self-enhancement values besides wealth. Some commented on how certain self-enhancement values would be inappropriate for researchers or could only serve as perverse incentives leading to scientific misconduct. Others thought of more science-specific meanings, such as satisfying a need for authority by leading research teams.

With regard to values missing from the original list of items, interviewees noted that the list seemed comprehensive, yet additional values emerged. Interviewees mentioned the following topics: 1) reasons for (going back to) doing research or choosing academia instead of another work environment, related to the intrinsic value of doing research, meritocracy, importance of relevant outcomes, and community; 2) a need for personal development, growth, learning, and having a good personal image; 3) being part of the research community and having social interactions with other researchers. A sense of belonging was covered in the values list but the interviewees framed this more specifically.

In sum, many personal values were deemed relevant for doing research. An initial outcome of this study is that the original wording of values can give rise to ambiguities in understanding items. This highlighted the need for further developing items after reevaluating the original set of values used here. If, for example, social recognition is used and defined in the same way as the original item (respect, approval by others), associations with academia will be less salient than if the description includes research-specific terms (e.g., respect and approval of the scientific community). If the values were rephrased, some value items could have separate meanings attached and added in more than one format. Another example is the original value related to honoring elders, that could be construed literally as honoring parents and family heritage, or could be rephrased for the academic context as an appreciation of senior researchers. For the next step, we refined the value items used in our interviews and developed a new list of values by eliminating items that the interviewees deemed irrelevant, adding the values they mentioned, and rewriting items with ambiguous meanings (see examples of interview quotes and transformation steps in S3 Appendix).

**4.2.3 Survey.** To further examine how personal values are understood in the context of science, we conducted a quantitative survey (part of a different study). Our aim was twofold: first, to further evaluate irrelevance and see if science-specific phrasings were deemed more relevant, especially in value domains associated with more ambiguous interpretations. Second, we wanted to get a sense of what participants deem as values in science compared to what researchers presumably value based on extant research [e.g., 4,16].

For this, we included in the survey a set of 60 values (Fig 4) after prompting responses to the following questions: "What do scientists value? How important do you think the following values are for scientists? Think about values that you think scientists value, values that are important for doing science." We asked participants to respond using a 7-point Likert scale

similar to those often used in value scales, ranging from 1- Extremely unlikely (for a value that could motivate some researchers but is unlikely to be important for most), to 7- Extremely likely (for values likely to be important for most). Participants could also indicate if they thought a value was completely irrelevant for researchers.

We opted to use the convenience sample of PhD candidates invited to take part in a larger survey conducted from 23-11-2020 to 1-2-2021. The exact recruitment and participation steps for this survey study are described in the project's OSF respository. The project was registered in the study proposal and ethical form approved by the Eindhoven University of Technology (TU/e) Ethical Review Board under ID: 1209. Participants provided written informed consent. We attached our value questions as a final section and informed participants that this section was optional and not related to the previous sets of questions. As we received sufficient responses to this optional part of the study, we discuss the results here.

From the complete sample size of the longer survey study ($N = 391$), a total of $N = 255$ PhD candidates (36% women, less than 1% gender variant/non-conforming) responded to the section on values. Respondents' average age was 28.8 years old, 45% of the participants were Dutch, 34% indicated belonging to an ethnic minority, and 24% to a racial minority. As the sample comprised PhD students from Eindhoven University of Technology, most respondents indicated technology as their main research area (58%), and fewer respondents engage in physical sciences (19%), social sciences (10%), life and biomedical sciences (10%), and arts and humanities (4%). The rate of perceptions about values as a factor of assigned importance and irrelevance is shown in Fig 4. Fig 5 presents value domains (in order) based on frequency perceptions.

In line with the literature review and the interviews, achievement, self-direction, and some benevolence and universalism values were rated highly, while most values relating to tradition were rated low in terms of their importance to researchers. However, most participants often deemed values rephrased to suit the scientific context and relating to hedonism or security at least more or less likely important to researchers. Outcomes regarding self-enhancement

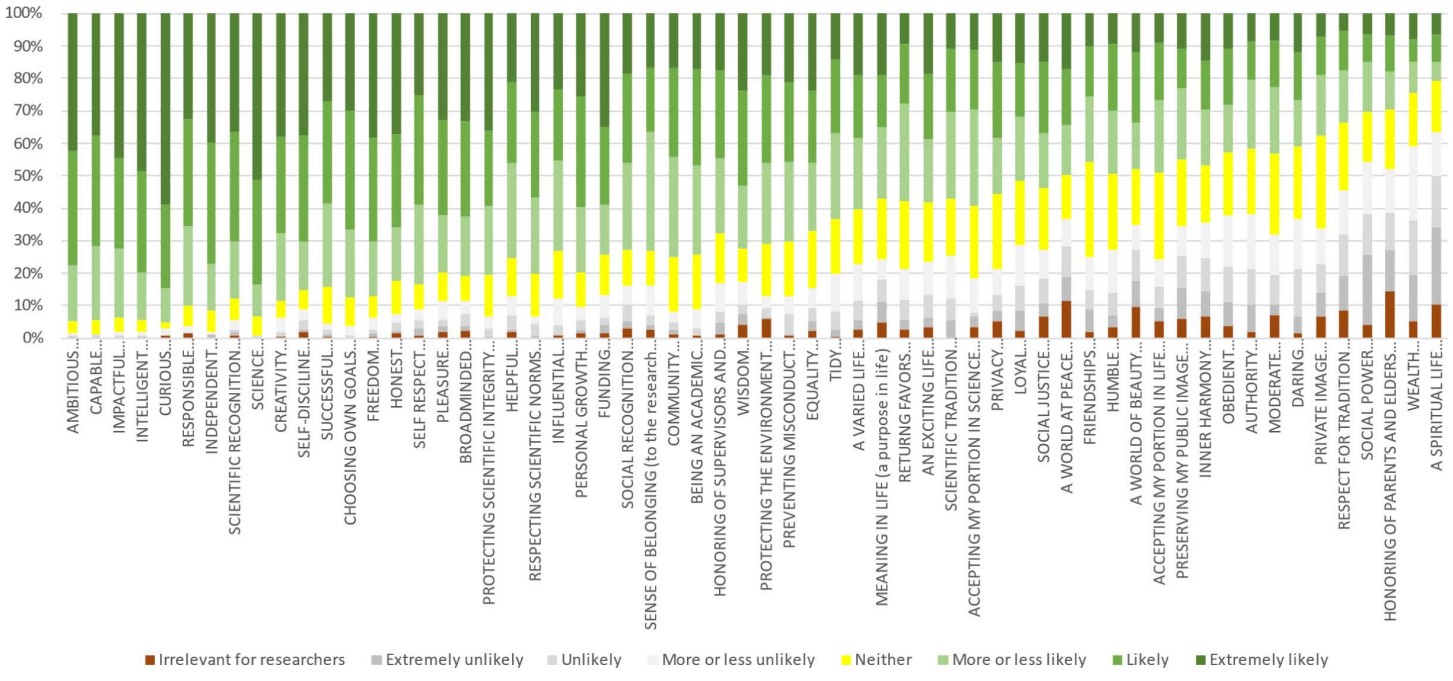

**Fig 4. Perceptions about the importance of values.**

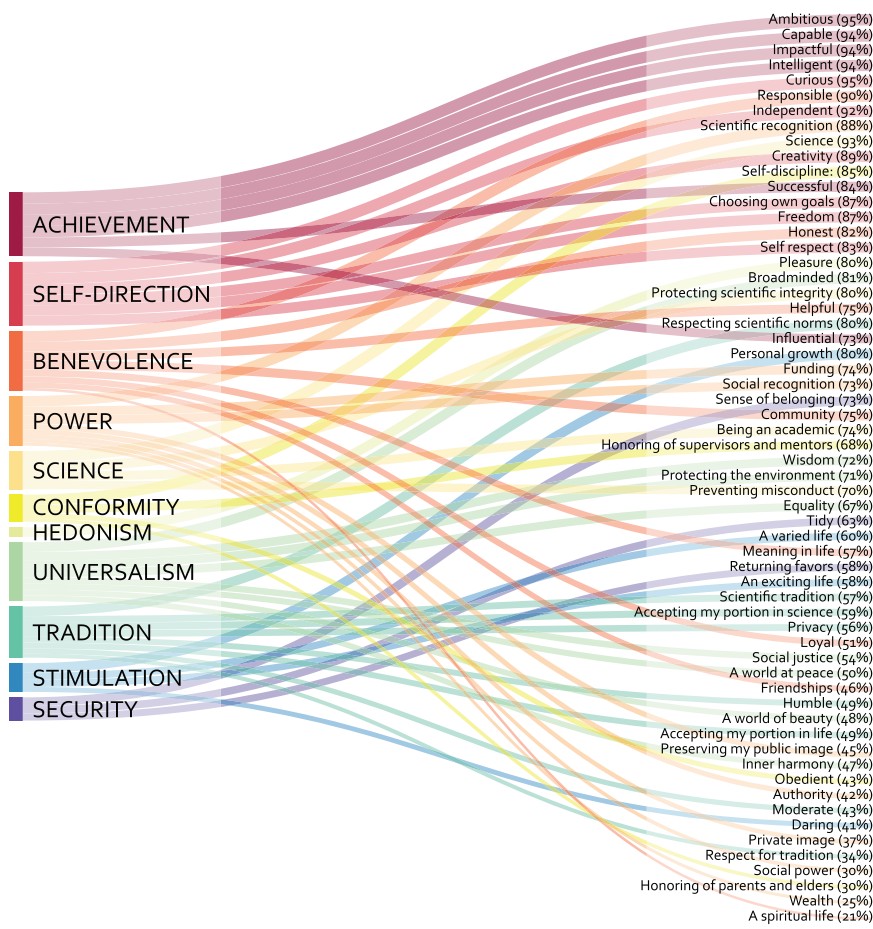

**Fig 5. Value domains (in order) as a factor of the rate participants respond at least "more or less likely" and "irrelevant" to connected value items (order based on Fig 4).**

values were ambivalent, but an underlying logic emerged: generic versions of power values (e.g., wealth, social recognition, or social power) and some achievement values (e.g., influential) scored much lower than their science-specific versions (e.g., funding, scientific recognition, or impactful). Unlike the original items, some science-specific versions of power and achievement values were rated important by the vast majority of participants.

This difference in the perceived relevance of generic and science-specific power, and achievement values in particular, questions the current portrayal of the values researchers deem important. While extant value research seems to explicitly [e.g., 16] or implicitly [e.g., 4] suggest that researchers are less concerned about values related to pleasurable experiences or reflecting ambitions for social or material influence, our outcomes suggest that science-specific distinctions can paint a very different picture of researchers and the relevant values in research. Given this outcome, we decided to reevaluate our approach.

## 4.3. Conceptualization

Based on our approach of fusing personal and work values and the results outlined above, we updated the preliminary definitions and delimitations of the construct and dimensions.

**4.3.1. Definition.** We define academic research values as "principles which serve as a basis of evaluating outcomes of scientific work-related actions, guide the selection of

scientific work goals, and represent the relative importance assigned to various academic job aspects related to research activities. They serve as guiding principles for a researcher or group of researchers' decisions in the academic work setting and are less broad than personal values, but still represent motivational goals that transcend specific work situations. Similar to personal values, academic research values are desirable in the sense that they represent important and worthy causes to researchers." Throughout this process, we have used the term scientific work values, in line with the literature terminology. However, after conceptualization, we decided the term "academic research values" better reflects our delimitation of the construct (i.e., researchers working in an academic setting).

**4.3.2. Dimensions.** We created final dimensions aligned with the value categories incorporated in our measurement review (Fig 6 and S4 Appendix).

**4.3.3. Item characteristics.** Based on a summary of all the sources reviewed so far and the redefinition of our construct, we decided on item characteristics (i.e., wording, form, and response types), as described in the next sections. For the final step, we generated a set of **246 items, 11 dimensions, and 34 sub-themes** reflecting the outcomes of our inductive and deductive methods (Table 3).

We developed item proposals that experts can test for content validity and that cover all the identified dimensions and themes based on the data collected so far. We obtained this item pool by conducting three synchronous electronic brainstorming sessions (as recommended

**Fig 6. Main personal, work, and scientific work value categories relating to the final dimensions.**

**Table 3. The refined dimensions, definitions, themes, and examples of items.**

| Dimensions | Definitions | Themes | Example items |
|---|---|---|---|
| Ambition | Career success through demonstrating competence according to academic standards, feelings of achievement and being a competent researcher | Career, Competence, Achievement | To win grants, scholarships, and scientific awards<br>To be highly cited<br>To believe in my own value as a researcher and feel self-respect |
| Authority | Scientific status, wealth, and prestige, control or dominance over other researchers and research resources, the importance of having a good public image as a researcher | Dominance over others, Dominance over resources, Influence, Prestige, Salary | To make decisions about who does what in a research project<br>To have direct influence over funding decisions<br>To have respect and attention for my research<br>To lead a prestigious research group<br>To know that my pay compares well with that of other workers |
| Autonomy | Freedom of thought and action: determination of work tasks, creating, and exploring own research topics | Freedom of thought/ Intellectual autonomy, Freedom of action/ Practical autonomy | To be able to set my own research agenda<br>To determine how I spend my workday |
| Benevolence | Being committed to the welfare of other researchers and emphasizing the importance of dependability and relationships within the research community | Caring for others, Dependability, Relationships | To help the people in my research community<br>To be on good terms with colleagues<br>To have good interactions with fellow researchers |
| Conformity | Conformity to scientific norms and codes of conduct, restraint of actions that might upset or harm others, abiding by social norms within the work environment | Scientific norms, Social norms, Codes of conduct | To work with researchers who respect scientific norms<br>To not speak up against more senior researchers<br>To return favors to collaborators and colleagues |
| Enjoyment | Seeking to take pleasure and gratification within the realm of scientific work, enjoying doing research | Pleasurable activities, Enjoying research | To go on nice conference trips<br>To take pleasure in the company of interesting, smart people<br>To enjoy my work |
| Organizational support | Fairness, support, and clarity within the research organization | Fairness, Support, Clarity | To know that the research institution handles processes fairly<br>To feel supported by the university I work at<br>To work in an environment in which norms and rules are clear |
| Tradition | Modesty about achievements and role as a researcher, respect, and acceptance of scientific traditions | Tradition, Modesty | To do scientific work which would be traditionally approved of<br>To be modest about my scientific achievements |
| Universalism | Assigning importance to research that has a positive social impact, sense of need to contribute to sustainability and prevent unethical or immoral research behaviors, and being tolerant to different approaches | Social impact, Sustainability, Tolerance, Research ethics | To better the world with my research<br>To make sure that the outcomes of my research do not have harmful consequences for nature<br>To be willing to consider other scientific perspectives<br>To protect scientific integrity |
| Variety | Being drawn to innovation, variety, novelty, and challenge in research, emphasizing the importance of personal growth and learning | Variety, Novelty, Challenge, Growth | To do varied work<br>To encounter exciting new ideas<br>To uncover hidden truths<br>To become the best researcher I can be |
| Working environment | Personal safety and comfort within the working and broader scientific environment, a sense of job security | Safety at work, Safety and wellbeing, Job security and stability | To work in an environment free from abusive relationships<br>To have well equipped infrastructure at my disposal (e.g., library, lab equipment)<br>To not be a subject of personal attacks for my research<br>To have a job that provides steady employment |

in Maaravi et al [51]) involving our team's five researchers, who were at various stages in their careers (a PhD candidate, an assistant professor, two associate professors, and a full professor) and had different disciplinary expertise. To cover a broader and more comprehensive set of topics than our theoretical view of the construct, as stated in best practice guidelines [30], we aimed to create a pool of 60 to 250 items. Undesirable items that might not be a good fit with the identified domains would be eliminated by successive evaluation.

To avoid construct underrepresentation, that is to say not capturing important aspects due to a narrow focus [30], we based our discussions on insights from all the previously mentioned sources as shown in Fig 7, and generated science-specific formats where possible. We did not add new dimensions unrelated to the literature and aimed to only include items based on value

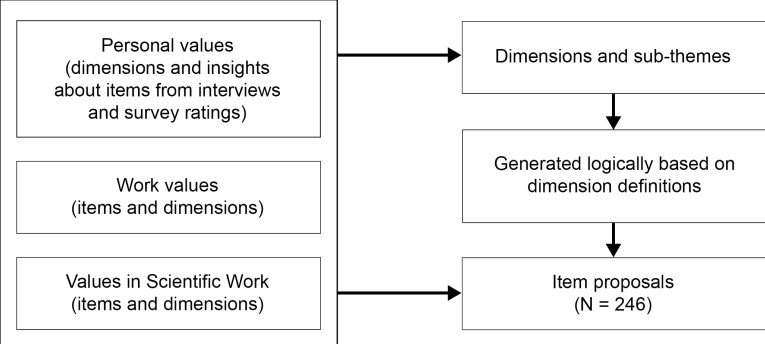

**Fig 7. Sources generating item proposals.**

dimension definitions to avoid construct-irrelevant variance. We excluded generic personal values such as "mature love" or "spiritual life" because of the interviewees' diverging interpretations, and survey participants' relevance ratings. During item generation, we grouped items around themes to provide a better overview of the types of values in each dimension.

We generated items based on three sources. Items were either: rephrased versions of the generic values based on our deductive methods; included as is or rephrased based on the value items in all the reviewed value instruments as listed in Section 4.2.1; or generated logically in line with the 11 dimensions' definitions. For example, based on the interviews and survey outcomes, we included a range of science-specific self-enhancement values not represented in English et al.'s Values in Scientific Work scale. This process resulted in a total of 246 item proposals spread over 11 dimensions and 34 themes. Details of our decision-making process for each item are available in S3 Appendix, and the final set of items is listed in S5 Appendix.

We implemented specific item wording (e.g., "To lead a prestigious research group") rather than generic phrasing of item proposals, to reduce ambiguity. Nuances of forming questions and items such as deciding between direct phrasing (thinking about values a person finds important: as a researcher, it is important to me…) and indirect phrasing (thinking about resemblance to a researcher with such values: how much is this researcher like you?), as well as response options (e.g., number of options on a Likert scale), require further methodological discussions.

Readability was checked by applying in Microsoft Word the Flesch Reading Ease (FRE) and Flesch-Kincaid (F-K) Grade Level formulas that have proven valid and reliable [52] to the entire list of value item proposals. Both scores (FRE: 64.3 and F-K grade level: 6.5) for reading difficulty indicated that the items were in plain or easy to read English and below college level.

We revisited the literature to account for important new studies since our initial review. At the start of our project, we found no comprehensive reviews of work value instruments nor an instrument based on the entire work values literature, whereas a recent study filled this gap by providing a summary of the most important work value domains and items in the Integrative Work Values Scale [17]. Although their instrument was developed in French, the authors gave definitions and an English example for each value domain in their scale. A comparison of their results and our value domains, themes, and items suggests we did not leave out any major work value aspects present in the literature (see S6 Appendix).

## 5. Discussion

This paper aimed to complement the discussion on the conceptualization and measurement of academic research values. To our knowledge, this is the first attempt at conceptualizing

academic research values based on an integrated perspective of the most broadly used personal and work value measurements. We integrated insights from the personal, work, and scientific work values literature as well as input from researchers, and presented the initial steps to develop value items for measurement purposes. At the end of this process, our conceptualization and initial items covered eleven dimensions of academic research values. These included value dimensions such as autonomy, variety, enjoyment, ambition, authority, conformity, tradition, benevolence, and universalism building on S. H. Schwartz's [13] framework and as implemented in the work context by Albrecht et al [24]. We added major work value measurements based on Leuty and Hansen [22], that tap into working environments and organizational support. We also incorporated value items presented in the Values in Scientific Work scale developed by English et al [4] and reevaluated our results using the so far most comprehensive review of the work values literature [17]. We designed scale items to be relevant and understandable for researchers working in an academic setting, regardless of discipline, career stage, or nationality, with a working level of English.

Our results suggest that for researchers, achievement, self-direction, and certain benevolence and universalism values are easily recognizable, even in their non-specific formulations. Most of our participants agreed on the importance of values associated with being ambitious, capable, intelligent, creative, independent, honest, responsible, or curious. Probably not a surprising result since many of these values are embedded within beliefs about the "positive" side of researchers' personality and motivations: intelligent, honest, curious, independent, see [53,54], or oft-cited within codes of conduct, research integrity courses, and discussions about the responsible conduct of research (honest, responsible) as well as prior measurements of scientific values [4,16,27]. However, the outcome of our research on less examined values seems relevant to several ongoing discussions.

Although the next phases of measurement development could still reshape our items, there are substantial differences between our values and others. Most notably, our set of items attaches more importance to ambition and authority values specific to scientific work in academia. Our results suggest that self-enhancement values phrased so they are easier to recognize in a scientific context such as having a scientific impact, are indeed deemed more relevant by researchers. By contrast, in their study of what individuals in various occupations value, Knafo and Sagiv [16] found no correlation between power or achievement value priorities and the investigative occupational environment, including research. The Values in Scientific Work scale developed by English et al [4] also dismissed the importance of ambition and authority for researchers. They included values emphasizing the importance of a good income but disregarded other potentially relevant self-enhancement values. Given our theoretical and empirical findings, especially from our survey, and an ongoing wider conversation about the importance of collaboration versus competition within science, we find it especially beneficial to study these values in the context of academic research.

Another unique feature of our approach is the integration of a comprehensive set of work values. While prior discussions about scientific values almost completely ignored the importance of safe, secure, and well-organized work environments, related needs are regarded as important factors, that many work-value scholars include in their instruments. This difference between the conceptualization of scientific and generic work values might simply stem from the lack of research specifically on scientific work values. It could, however, also reflect a disconnect between perceiving academic research as a career and a job, rather than (solely) a calling. Only discussing and prioritizing values connected to scientific norms and thought processes would largely disregard modern-day academics' lived experiences. Such experiences are embodied in a growing line of empirical and theoretical studies, opinion pieces, and non-peer reviewed sources discussing academic pressures, precarious or toxic work environments,

and scholars leaving academia on account of these institutional challenges (recent examples include: Kis et al.[55]; McKenzie [56]; Pelletier et al. [57]; Pruit et al. [58]; Skakni et al. [59]). This discussion underlines the importance of studying work values that reflect needs connected to safe, secure, and healthy work environments. In addition, these organizational characteristics can reflect a more diverse group of researchers' needs and promote the understanding of how we can increase the sustainability of academic career paths.

Our conceptualization of relevant value dimensions apparently aligns with a range of findings about researchers' motivations and personalities. For example, the value dimensions and their definitions we present here overlap with the motivational factors discussed in summaries by Johnson and Dieckmann [54] as well as Jussim et al [3]. Particularly in their review of motivations for doing scientific research, Johnson and Dieckmann list items such as assigning importance to making money, gaining power and fame, being liked and respected, being independent, doing good science, and helping society and others. In their Social Psychological Model of Scientific Practices, Jussim et al [3] include fame, job security, promotions, respect, and being paid well as researchers' personal motivations. While mostly not from a motivational perspective, the role of curiosity, creativity, and intelligence in scientific practice is also debated in detail in the psychology of science literature [60].

## 5.1. Benefits of studying academic research values

A current widely debated topic is what we should value in terms of the competitive versus collaborative nature of science. One recent example of this debate close to our team was when the president of the executive board of the Netherlands Organization for Scientific Research (NWO) compared science to elite sport and scientists to top athletes, competing for attention, impact, and new records [61]. Many Dutch researchers disagreed [62]. Levi's claims were classed as out of date and out of touch with the ongoing discussions about the harmful effects of competition within academia [63]. Indeed, though academic leaders often make such comparisons to sports or gaming [64], some researchers point out the harm caused due to the competitive nature of the academic reward system — and the narratives still promoting it [28,64–67]. In this line of research, competition is cited as harmful to diverse aspects of the responsible conduct of research, including integrity, credibility, reliability, openness, transparency, and cooperative knowledge generation.

Values can add to this discussion regarding the potential effects of competition in science. The values literature notes that self-enhancement values (power, achievement) influence competition and unethical behaviors, whereas self-transcendence values (benevolence, universalism) facilitate cooperation and prosocial behavior [7] and are negatively associated with unethicality [68]. In organizations, employees who assign a higher value to self-enhancement are more likely to compete and care more about status and prestige, while those who value self-transcendence are more likely to engage in altruistic behavior and cooperate rather than compete [69]. In addition, drawing attention to preferable values can facilitate behaviors consistent with the value in question [7,70].

Based on these results, academic research values could potentially be important drivers of behavior change. Influencing or activating certain values can lead to a change in associated behaviors. If we recognize which values influence ethical choices in research practices — behaviors on the spectrum ranging from ethical to unethical, including actions associated with responsible conduct of research as well as scientific misconduct and questionable research practices — we might be able to develop better behavioral change interventions to facilitate good and discourage bad practices. In a scientific context, Bruton et al [71] highlight institutional and career-oriented incentives such as competition underlying questionable research practices. And while codes of scientific conduct offer a range of virtues, norms, and values

supporting research integrity, they suffer from terminological challenges and an irreducible pluralism in what they prescribe [72], that makes understanding what is valued in science a complex and often cognitively demanding endeavor. Describing the initial steps to construct a measure can help the scientific community understand what researchers think is valued within science, linked to what they value as individuals in science. Such efforts can facilitate a simplified, honest conversation about what we as a community should or should not value in science or in a researcher.

Similarly to how researchers' personality traits help raise awareness of what role individual differences play in the research process [1], understanding their own values might give researchers insights into their internal psychological processes. Appreciating what researchers value and how their values influence their research-related behaviors can help them make more self-aware decisions and gain more control over their actions in the scientific process. Such understanding can be converted into greater awareness of the risk factors questionable practices, but might also reveal potential opportunities to find value-congruent ways of engaging in responsible conduct of research or good scientific citizenship.

Studies show that values explain a range of attitudes and behaviors in the work environment. As summarized by English et al [4], individuals tend to choose and remain more satisfied in jobs that align with their values. If employees' values match those of their working environment, they are more satisfied with their jobs and more likely to commit to, identify and stay with the organization [69,73]. Value congruence is beneficial for employees' subjective well-being and can also benefit employee performance [69]. Value misfit with the organization can leave employees feeling out of place, stressed, and unfulfilled by their work, consequently leading to lower engagement, performance, organizational commitment, and a greater desire to quit, as well as higher employee turnover (as outlined by [4,24,74]). In the context of academic research values, this could translate into recognizing misfits between the values central to researchers seeking cooperation (building collaborative science) and the values they perceive as reinforced or rewarded within science. Understanding the career effects of such value (in)congruence will help us see to what extent valuing competition might be detrimental to some researchers' careers. We might also be able to increase their work satisfaction, well-being, and productivity by providing incentives aligned with their values.

## 5.2. Limitations and future research

This study provided an initial set of academic research value items that can be used for content validity testing and evaluation of psychometric properties. Our outcomes seem comprehensive when measured against the sources we incorporated. For our research, however, we only included the personal, work, and academic research value literature that defines values as psychological constructs. We also excluded many instruments measuring such values and only relied on widely used ones, selected on account of the combination of validity, relevance, and connectedness (between different lines of the value literature). Compared to the most exhaustive work values instrument introduced up till now (see S6 Appendix) and the overlap between our value dimensions and the literature on researchers' motivations discussed above, our results seem to be comprehensive.

While we did try to limit the arbitrary nature of our choices and include empirical results following best practice recommendations, the convenience samples for our inductive methods are a further limitation. We based our decisions on PhD candidates' perceptions. This reliance on a non-representative, relatively small, and in many aspects homogeneous sample could have biased our approach. However, this sampling strategy was justified both by our aim to develop a scale of value items that even the least experienced researchers can relate to, and the intended outcome of these initial investigations: gain a preliminary understanding of the

specificity required for measuring our construct. Future research steps related to item selection and validation will need to involve a more diversified sample of the target population. Whether these choices result in a comprehensive and valid set of items will be tested in the ensuing phases of measure development.

Variations in organizational cultures is a connected concern, potentially further limiting the generalizability of our findings. Organizational culture consists of a set of values, beliefs, and assumptions that guide employees within an organization. As the cultural values of organizations are influenced by national, occupational, industrial, and individual values [75], academic institutions across the globe can be expected to prescribe different sets of values for their employees. This can lead to differences in academics' research values as well, potentially decreasing the universality of our scale. For example, the cash-per-publication monetary reward system of science in China [76] isa very different reward structure than that of the Netherlands. While in our sample we found that values reflecting direct monetary incentives were less accepted, this finding might not generalize to an organizational culture that inherently values cash rewards.

It can also be expected that organizations would differ in terms of competitiveness or hierarchical structures, leading to differences in self-enhancement values for example. The fact that there is a lack of universality in the set of values in our study is not a problem by itself. Some dimensions and values might apply differently to different researcher groups, leading to differences that can help us learn about the underlying reasons (e.g., different national or organizational cultures). However, we might indeed be missing some values that were not recognized in our sample. This limitation can be dealt with similarly to how the generalizability of personal values evolved in the past decades: by repeating this study with a larger, more diverse set of participants and continuing discussions about the values of researchers.

Further research is necessary to confirm the external and internal validity of our instrument. The specificity of our construct as contrasted by Schwartz's universal personal values also poses the question whether academic research values can be universally applicable to all academic contexts. While the values discussed in this paper may appear specific, they are rooted in well-researched personal and work values that have established their generalizability based on a research line spanning decades [see for example: 77, 78]. As such, they likely reflect a broader, more general set of values that are common among researchers, albeit expressed differently depending on the context. The assumption that researchers across disciplines, cultures, or age groups would possess entirely different values seems unlikely; rather, it is more plausible that the values we identified are present to varying degrees in different environments.

After thorough validation, the scale could be used to generate in-depth knowledge about specific subsets of values relevant in certain contexts or for particular purposes (e.g., self-enhancement versus self-transcendence values in connection with competition versus cooperation in science). One of the core reasons for studying values was to explore their relationships with real-life behaviors. Future research could investigate causal relationships through experimental manipulations. Ensuring both external and internal validity will be crucial for the scale's applicability and reliability in diverse settings.

## 6. Conclusion

Further research and discussion are needed to reach an agreement on what researchers value and before an academic research values scale can become a valid and reliable instrument. As outlined above, the next step is content validation, ideally involving experts and the target population. Once a pre-validated set of items emerges, the next steps in the measure

development process can begin, according to best practice recommendations [30]. Due to the complexity of this construct, we anticipate that future validation efforts will take place on sub-scales, building up the entire scale over time. Full scale validation is an ambitious next step, especially if the aim is to represent values relating to a diverse set of researchers across a broad range of characteristics including discipline, academic status, age, gender, nationality and ethnicity, culture, and so on. While such efforts will be labor-intensive and expensive, the return on this investment seems worthwhile for funders.

## Supporting information

**S1 Appendix. Value Instruments.**
(XLSX)

**S2 Appendix. Dimension Definitions.**
(XLSX)

**S3 Appendix. Value Development.**
(XLSX)

**S4 Appendix. Development of our Dimensions and Sub-themes.**
(XLSX)

**S5 Appendix. List of Value Item Proposals.**
(XLSX)

**S6 Appendix. Comparison with the Integrative Work Values Scale.**
(XLSX)

**S7 Appendix. Figure 2 Table version.**
(XLSX)

**S8 Appendix. Overview of Appendices.**
(XLSX)

## Acknowledgements

A special thanks to Val Kidd for copy editing the manuscript.

## Author contributions

**Conceptualization:** Andrea Kis, Elena M. Tur, Krist Vaesen, Wybo Houkes, Daniel Lakens.

**Data curation:** Andrea Kis.

**Formal analysis:** Andrea Kis.

**Funding acquisition:** Elena M. Tur, Wybo Houkes, Daniel Lakens.

**Investigation:** Andrea Kis.

**Methodology:** Andrea Kis, Daniel Lakens.

**Project administration:** Andrea Kis.

**Supervision:** Elena M. Tur, Krist Vaesen, Wybo Houkes, Daniel Lakens.

**Visualization:** Andrea Kis.

**Writing – original draft:** Andrea Kis.

**Writing – review & editing:** Andrea Kis, Elena M. Tur, Krist Vaesen, Wybo Houkes, Daniel Lakens.

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
