## [Decision Letter · Decision Letter 0]

25 Jul 2024

PONE-D-23-39962Academic research values: Conceptualization and initial steps of scale developmentPLOS ONE

Dear Dr. Kis,

Thank you for submitting your manuscript to PLOS ONE. After careful consideration, we feel that it has merit but does not fully meet PLOS ONE’s publication criteria as it currently stands. Therefore, we invite you to submit a revised version of the manuscript that addresses the points raised during the review process.

We look forward to receiving your revised manuscript.

Kind regards,

Myriam M. Altamirano-Bustamante

Academic Editor

PLOS ONE

Reviewers' comments:

Reviewer's Responses to Questions

**Comments to the Author**

1. Is the manuscript technically sound, and do the data support the conclusions?

Reviewer #1: Yes

Reviewer #2: Partly

2. Has the statistical analysis been performed appropriately and rigorously? 

Reviewer #1: N/A

Reviewer #2: N/A

3. Have the authors made all data underlying the findings in their manuscript fully available?

Reviewer #1: Yes

Reviewer #2: Yes

4. Is the manuscript presented in an intelligible fashion and written in standard English?

Reviewer #1: Yes

Reviewer #2: Yes

5. Review Comments to the Author

Reviewer #1: This is a fascinating and much-needed piece. I am pleased to read other authors exploring the values that direct decisions and behaviors in science. I find it essential to have more. However, I would suggest several things to improve the readability of the paper:

1. This type of study follows a very complex novel methodology; supporting it with a visual diagram/roadmap can help the readers better understand what you did and when you did it. Please include one figure where you show this. It will embellish your paper a lot and will clarify any possible confusion that the reader may be getting from your written methodology. Make sure you keep your methods consistent with the image.

2. I believe sections 1 and 2 could be merged into a single introduction section. While the content is good, I would enjoy a more concise version. Make sure you state your research question, objectives, and hypotheses right before the methodology. I see that you did it at the end of section 1. Still, it makes it easier for the reader to collect their thoughts and understand why they read what they read in the introduction while preparing them for the methodology (which you will summarize in a diagram, too).

3. Talking about values is like diving into murky waters. There are so many definitions of values, and you used a behavioral framework for your work. While I understand that is cleaner than diving into the more philosophical and ethical aspects, it would complete your paper to dedicate just one or two paragraphs to ethics. Please do not think I want you to talk about what it means to be good or bad, what actions are reprehensible, and so on - remember that ethics studies moral judgments and these judgments require values. What is a moral judgment? If you have a set of actions and a set that contains good and bad, a moral judgment is a relation that assigns each action the value of good or bad (an arrow that links each action to either "good" or "bad" if you may). The chosen relation (the arrow you choose) is a function of values. You can check Hartmann's definition for simplicity, but there are others. You can even find that Thomas Kuhn defines values in the context of science!

4. On that same note, check Javier Echeverría's theory on value systems in science and technology. Add it to your paragraph on ethics and use it to enrich your discussions, which are already pretty good.

5. Do not forget to talk about organizational cultures!

6. I strongly recommend you have graphic/visual support for the conclusion. Where did you arrive? What are the following destinations?

Finally, I would like to trigger further discussions for you to choose whether you want to include them in your paper. Do you believe a single psychometric instrument can circumscribe the universe of values? You briefly discuss this in the study's limitations, but values are elusive; they often arise entangled with other values and do not always have the same exact meaning in all organizational cultures! Would you maybe want to study a subset of specific values? If so, why would you study these and not other values? What do you think of qualitative research as a way to outline values in scientific practice? Can we expect to have external validity in such studies?

Once again, this is already a pretty good article that needs some polishing. Please ensure it is concise and takes readers from your research question to your findings. Support it with good artwork and add more on philosophy, do not forget to enrich your discussion with some of the suggestions above if you see fit.

Reviewer #2: The manuscript titled “Academic research values: Conceptualization and initial steps of scale development” presents an engaging study on the role of values and how they affect the decisions of researchers, but also sets the first steps what the authors believe to be the initial steps in measuring those values. The goal of this study is to analyze conceptually the values specific to scientific work and to empirically develop measurements to be able to know which values are present in scientific communities. The authors rightly claim that advancing the empirical knowledge of these values would have an impact on scientific careers allowing to impact on the outcomes, in the interest shown by young scientists and also who to avoid questionable practices.

Conceptually the authors seek through a literature review scales that are able to measure values, they integrate previous work done on personal values and work values. Much of the study is centered around the contributions of S.H. Schwartz in personal values that are considered as the principles that guide actions of a person. The four dimensions on which Schwartz maps personal values (openness to change, self-transcendence, conservatism and self-enhancement) tend to be very universal and they are able to elicit other values that explain how individuals act from them. How ever these values are quite removed from some aspects of the work environment, so the authors integrate to their concept of academic values those related to work such as work-satisfaction, engagement, career adaptability and others, as found in the work of Busque-Carrier et al. But still scientific work is not a regular kind of work, so they still integrate scientific values, where autonomy, social impact, innovation are important and not considered before. To cover these values they rely on the work of English et al.; and they are analytically integrated to match these more specific values with those basic in Schwartz conception.

Methodologically the authors strive for content validity of these values in a three-step study, the conceptual construction just described was based on a literature review, and afterwards these values were tested against the interviews made to researchers and a more ample survey. As part of the results a more refined conception of the values was elicited, for example, some of the values conceptually reached were deemed completely irrelevant; on the contrary other values were specified extensively to the context of scientific work values.

The discussion is very interesting and takes care of the details of how to encompass the three types of value and articulate them into the first steps of a specific scale concerning scientific work values. In this discussion also the subject of the benefits of studying research values, because a better understanding of scientific practices may allow to influence which of them should be privileged in order to accomplish better results in research, but also in the attitudes and environments of the scientific community.

Considering the remarks I have made, I would like to point out some observations I have concerning the manuscript:

1. The methods section in all three stages of the study a wider attention should be given to the description of the followed steps: a) in the case of the literature search the reader does not know the period of the search, or how large were the results, as well as the specific criteria to reject or accept the papers that were chose. b) In the case of the interviews the demographic information of the participants would be insightful; and c) in the case of the survey the authors mention it as a problem in their limitations section (pg. 28) that the surveys are made to PhD candidates, who may not yet be considered to be aware of the ins and outs of scientific research. In this last regard I do not see how to work around this limitation.

2. Figure 2 has some problems for its visualization, its wording is in some of the boxes incomplete.

3. My last remark has to do with grounding that this work has on the paper by Schwartz: the conception of personal values is very general and can be considered universal without any problems, however when moving to more specific work related values, and later on, scientific work values there are very specific values. The authors acknowledge that this could be the case in different disciplines, ages or cultures, hence, bit more could be said in their conceptual analysis on why it is suitable for these values that are very specific to scientific practice to still be considered universally in different scientific communities.

6. PLOS authors have the option to publish the peer review history of their article (what does this mean? ). If published, this will include your full peer review and any attached files.

**Do you want your identity to be public for this peer review?** For information about this choice, including consent withdrawal, please see our Privacy Policy .

Reviewer #1: No

Reviewer #2: No

---

## [Author Response · Author response to Decision Letter 0]

6 Sep 2024

Subject: Response to Reviewer Comments for Manuscript PONE-D-23-39962: Academic research values: Conceptualization and initial steps of scale development

Dear Myriam M. Altamirano-Bustamante,

Thank you for the opportunity to revise and resubmit our manuscript, titled “Academic research values: Conceptualization and initial steps of scale development.” We appreciate the valuable feedback provided by the reviewers, which has been instrumental in refining our manuscript. We understand that the complexity of the issues and methodology discussed in our paper may present challenges in readability. In response to the reviewers' comments, we carefully considered ways to streamline the content and enhance clarity.

During the revision process, we explored various options, including visualizations and additional explanatory details. After thorough deliberation, we reduced the amount of extra information added to the paper to maintain the focus and coherence of the manuscript. Our aim was to balance thoroughness with readability, ensuring that the core arguments and findings remain clear and accessible to the audience.

Below, we provide a detailed point-by-point response to the reviewers' comments, outlining the specific revisions we have made and the rationale behind our decisions. For clarity, the reviewers' comments are included, followed by our responses and the corresponding changes made to the manuscript. All manuscript text changes are marked in the revised manuscript document and highlighted in this letter.

Reviewer 1:

Comment 1:

This type of study follows a very complex novel methodology; supporting it with a visual diagram/roadmap can help the readers better understand what you did and when you did it. Please include one figure where you show this. It will embellish your paper a lot and will clarify any possible confusion that the reader may be getting from your written methodology. Make sure you keep your methods consistent with the image.

Response: Thank you for this suggestion, we agree that a diagram increases the coherence and clarity of the paper. To visualize the methodological steps, our conclusion, and the future research that is needed, we have created a figure (Figure 1).

Figure 1.

Overview of the scale development process, results, and future research (in brackets: the year of carrying out the activity)

Comment 2:

I believe sections 1 and 2 could be merged into a single introduction section. While the content is good, I would enjoy a more concise version. Make sure you state your research question, objectives, and hypotheses right before the methodology. I see that you did it at the end of section 1. Still, it makes it easier for the reader to collect their thoughts and understand why they read what they read in the introduction while preparing them for the methodology (which you will summarize in a diagram, too).

Response:

Thank you for highlighting these important points. We agree that the conciseness of the first part of the paper could be improved. However, we believe that keeping the two sections separate enhances the overall readability and clarity of the paper. To address your concern, we have revised the introduction to provide a clearer overview of the paper's structure and emphasized the distinct purposes of each section. The introduction now focuses on introducing the topic and outlining the content of the paper, while the literature review is dedicated to grounding our work within the relevant theoretical frameworks. We hope these revisions improve the flow and clarity of the manuscript.

To predict research-related behaviors, we aimed to develop a scale to measure researchers’ psychological values. While values are broadly used constructs across various disciplines, our approach focuses specifically on psychological values. The current study is rooted in research that defines values as underlying psychological criteria guiding behaviors and preferences. This focus is due to our goal of describing individuals’ values and connected behaviors, necessitating a construct capable of achieving this. Psychological values are widely acknowledged for their ability to provide a descriptive, rather than normative, approach to understanding people’s motivations and goals. Consequently, our approach excludes broader definitions of values, such as core universal moral values [8], scientific virtues [9], and scientific values [10].

In the remainder of this paper, we present these scale development steps. First, to ground our work, we discuss the personal, work, and scientific work value literature. Next, we present our methods, starting with the concept development steps that serve as precursors of developing a conceptually sound, comprehensive description of psychological values relevant to researchers in a scientific context. Then, we highlight the item generation process through which we delimited and finalized our definitions and through which we created an initial set of scientific work values that can form the basis for future measure development work (see the process summarized in Fig 1).

Fig 1. Overview of the scale development process, results, and future research (in brackets: the year of carrying out the activity)

Comment 3:

Talking about values is like diving into murky waters. There are so many definitions of values, and you used a behavioral framework for your work. While I understand that is cleaner than diving into the more philosophical and ethical aspects, it would complete your paper to dedicate just one or two paragraphs to ethics. Please do not think I want you to talk about what it means to be good or bad, what actions are reprehensible, and so on - remember that ethics studies moral judgments and these judgments require values. What is a moral judgment? If you have a set of actions and a set that contains good and bad, a moral judgment is a relation that assigns each action the value of good or bad (an arrow that links each action to either "good" or "bad" if you may). The chosen relation (the arrow you choose) is a function of values. You can check Hartmann's definition for simplicity, but there are others. You can even find that Thomas Kuhn defines values in the context of science!

Response:

We appreciate the enthusiastic response and engaging ideas of the reviewer and have carefully considered these points. In an earlier version of the paper we also tried to reflect more on the different aspects of moral values and judgements, virtues, and other connected ethical considerations. Unfortunately it made the paper less readable and concise in the end. Our conclusion then was that values in different scientific disciplines are elusive constructs, making it difficult to integrate an all-encompassing overview without further complicating the text. To increase readability and coherence, but also because of the distinct defining features of psychological values, we decided to focus on the psychological approach instead of a broader perspective. The explicit choice we made about basing our work on Schwartz’s theory and excluding all non-psychological value theories when designing our methodology also aided this decision, providing us with a clear distinction between psychological and non-psychological values.

Comment 4:

On that same note, check Javier Echeverría's theory on value systems in science and technology. Add it to your paragraph on ethics and use it to enrich your discussions, which are already pretty good.

Response:

Thank you for this additional comment. We considered it but arrived at the same conclusions we discussed as a response to Comment 3.

Comment 5:

Do not forget to talk about organizational cultures!

Response:

We agree that organizational cultures are very relevant to this discussion. A reflection on their interconnectedness with values is now included in the limitations section:

Variations in organizational cultures is a connected concern, potentially further limiting the generalizability of our findings. Organizational culture consists of a set of values, beliefs, and assumptions that guide employees within an organization. As the cultural values of organizations are influenced by national, occupational, industrial, and individual values [75], academic institutions across the globe can be expected to prescribe different sets of values for their employees. This can lead to differences in academics’ research values as well, potentially decreasing the universality of our scale. For example, the cash-per-publication monetary reward system of science in China [76] isa very different reward structure than that of the Netherlands. While in our sample we found that values reflecting direct monetary incentives were less accepted, this finding might not generalize to an organizational culture that inherently values cash rewards.

It can also be expected that organizations would differ in terms of competitiveness or hierarchical structures, leading to differences in self-enhancement values for example. The fact that there is a lack of universality in the set of values in our study is not a problem by itself. Some dimensions and values might apply differently to different researcher groups, leading to differences that can help us learn about the underlying reasons (e.g., different national or organizational cultures). However, we might indeed be missing some values that were not recognized in our sample. This limitation can be dealt with similarly to how the generalizability of personal values evolved in the past decades: by repeating this study with a larger, more diverse set of participants and continuing discussions about the values of researchers.

Comment 6:

I strongly recommend you have graphic/visual support for the conclusion. Where did you arrive? What are the following destinations?

Response:

Thank you for this suggestion, we integrated this information (i.e., both where we arrived with our results as well as the intended future research paths) into Figure 1.

Comment 7:

Finally, I would like to trigger further discussions for you to choose whether you want to include them in your paper. Do you believe a single psychometric instrument can circumscribe the universe of values? You briefly discuss this in the study's limitations, but values are elusive; they often arise entangled with other values and do not always have the same exact meaning in all organizational cultures! Would you maybe want to study a subset of specific values? If so, why would you study these and not other values? What do you think of qualitative research as a way to outline values in scientific practice? Can we expect to have external validity in such studies?

Response:

We added the following section to the paper to reflect on validity and the universality of values:

Further research is necessary to confirm the external and internal validity of our instrument. The specificity of our construct as contrasted by Schwartz’s universal personal values also poses the question whether academic research values can be universally applicable to all academic contexts. While the values discussed in this paper may appear specific, they are rooted in well-researched personal and work values that have established their generalizability based on a research line spanning decades [see for example: 77, 78] . As such, they likely reflect a broader, more general set of values that are common among researchers, albeit expressed differently depending on the context. The assumption that researchers across disciplines, cultures, or age groups would possess entirely different values seems unlikely; rather, it is more plausible that the values we identified are present to varying degrees in different environments.

After thorough validation, the scale could be used to generate in-depth knowledge about specific subsets of values relevant in certain contexts or for particular purposes (e.g., self-enhancement versus self-transcendence values in connection with competition versus cooperation in science). One of the core reasons for studying values was to explore their relationships with real-life behaviors. Future research could investigate causal relationships through experimental manipulations. Ensuring both external and internal validity will be crucial for the scale’s applicability and reliability in diverse settings.

Reviewer 2:

Comment 1:

The methods section in all three stages of the study a wider attention should be given to the description of the followed steps: a) in the case of the literature search the reader does not know the period of the search, or how large were the results, as well as the specific criteria to reject or accept the papers that were chose. b) In the case of the interviews the demographic information of the participants would be insightful; and c) in the case of the survey the authors mention it as a problem in their limitations section (pg. 28) that the surveys are made to PhD candidates, who may not yet be considered to be aware of the ins and outs of scientific research. In this last regard I do not see how to work around this limitation.

Response:

a) We updated the literature search methodology with additional information:

A literature search conducted from 2020 January to May identified instruments that measure relevant values for scientific work. We sought reviews of validated measures of work values using these search terms: work value combined with scale, measur*, instrument, inventory, questionnaire, model, validity or validat* in the Web of Science, APA PsycArticles, APA PsycInfo, ERIC, and SCOPUS databases. Given the scope and aim of the review, no restrictions were placed on the publication period. The search yielded a substantial number of articles (3027 results in total, 1941 results after excluding duplicates), which were then screened for relevance based on titles and abstracts. Full-text articles were reviewed to determine their alignment with the research objectives. We only included studies conceptualizing work values as psychological constructs and excluded constructs not in line with our preliminary value definition, for example, social work values, occupational work values, and work ethic values (i.e., Confucian, Islamic, protestant). Because synthesizing the large number of different conceptualizations of work values was beyond the scope of our research, we further limited our search to sources that provide summaries or comparisons of existing and widely used scales.

b) We added the demographic information collected from our participants, as follows:

We used a convenience sample consisting of six PhD candidates (university employees in the Netherlands, four internationals and two Dutch nationals, four women and two men, five in their first PhD year and one in their final year).

c) We thank the reviewer for their thoughtful comment and agree that this limitation posed by our sample is difficult to overcome. Still, as it poses a concern to the interpretation of our results, it seemed relevant to briefly discuss in the limitations section of the paper.

Comment 2:

Figure 2 has some problems for its visualization, its wording is incomplete in some of the boxes.

Response:

Thank you for noticing these errors. We corrected the figure with the complete wording in each box.

Comment 3:

My last remark has to do with grounding that this work has on the paper by Schwartz: the conception of personal values is very general and can be considered universal without any problems, however when moving to more specific work related values, and later on, scientific work values there are very specific values. The authors acknowledge that this could be the case in different disciplines, ages or cultures, hence, bit more could be said in their conceptual analysis on why it is suitable for these values that are very specific to scientific practice to still be considered universally in different scientific communities.

Response:

Thank you for your insightful remark regarding the universality of scientific work values as derived from Schwartz's general conception of personal values. We appreciate the opportunity to further clarify our position on this matter.

While we acknowledge that scientific work values may appear sp

---

## [Editor Report · Decision Letter 1]

10 Jan 2025

Academic research values: Conceptualization and initial steps of scale development

PONE-D-23-39962R1

Dear Dr. Kis

We’re pleased to inform you that your manuscript has been judged scientifically suitable for publication and will be formally accepted for publication once it meets all outstanding technical requirements.

Kind regards,

Myriam M. Altamirano-Bustamante

Academic Editor

PLOS ONE
---

## [Editor Report · Acceptance letter]

PONE-D-23-39962R1

PLOS ONE

Dear Dr. Kis,

I'm pleased to inform you that your manuscript has been deemed suitable for publication in PLOS ONE. Congratulations! Your manuscript is now being handed over to our production team.

Kind regards,

on behalf of

Dr. Myriam M. Altamirano-Bustamante

Academic Editor

PLOS ONE